# Association between Hpa Axis Functioning and Mental Health in Maltreated Children and Adolescents: A Systematic Literature Review

**DOI:** 10.3390/children10081344

**Published:** 2023-08-03

**Authors:** Pierre-Antoine Leroux, Nolwenn Dissaux, Jean Yves Le Reste, Guillaume Bronsard, Nathalie Lavenne-Collot

**Affiliations:** 1Service de Psychiatrie de l’Enfant et de l’Adolescent, CHRU, 29200 Brest, France; 2Faculté de Médecine, Université de Bretagne Occidentale, 29200 Brest, France; 3EA 7479, 29200 Brest, France; 4Département de Sciences Humaines et Sociales, EA 3279 (CEReSS, AMU), 29200 Brest, France; 5Laboratoire du Traitement de l’Information Médicale, Inserm U1101, 29200 Brest, France

**Keywords:** childhood, hypothalamo hypophyseal system, cortisol, physiological stress reactivity, behavior disorders

## Abstract

Background: Previous studies have demonstrated that children who experience maltreatment show a more elevated risk of psychopathological disorders than children from the general population. The HPA (hypothalamic–pituitary–adrenal) axis is not mature at birth and undergoes strong social regulation during the first years of life. Consequently, early exposure to stress could modify the usual adaptative response to stress. In stressful situations, perturbations in both cortisol response and cortisol circadian rhythm have been observed. Nevertheless, studies that have evaluated the links between child abuse, dysregulation of the HPA axis, and mental disorders have shown diverse results. Because of the variety of methods employed in the different studies, no formal comparisons have been made. In this systematic review, we have brought together these results. Methods: We conducted a systematic review of studies analyzing the correlation between child abuse, mental disorders, and HPA axis activity in patients aged between 6 and 16 years. PubMed, Scopus, Cochrane, and Google Scholar were searched using relevant keywords and inclusion/exclusion criteria (from 2000 to 2020). Results: Fifteen studies from the 351 identified were included. Most patients were children in the child welfare system. Children who had experienced child abuse presented with more severe mental disorders (particularly in the dimensional measure) than children who had not been abused. HPA axis activity was assessed by measuring basal cortisol for some studies and cortisol reactivity for other studies. For children experiencing child abuse, there was a possible association between abuse and a decrease in the reactivity of the HPA axis. In addition, early life stress could be associated with lower matinal cortisol. However, the association between mental disorders and cortisol secretion in maltreated children did not seem obvious. Conclusions: This systematic review demonstrates that mental disorders are more frequent and severe in cases where child abuse has occurred. Moreover, children who experienced child abuse seem to present changes in the reactivity of the HPA axis. Nevertheless, the potential correlation between these changes in the reactivity of the HPA axis and mental disorders in this population needs to be evaluated in further studies.

## 1. Introduction

In non-human primates, disturbances in early maternal–infant interactions are important factors in the risk of developing psychopathological disorders [1].

In children, the influence of the environment on development and psychopathology has been known for a long time, for example, in Bowlby’s Theory of Attachment, which highlights the central role of parent–child attachment in the child’s psychological development [2]. Children exposed to early maltreatment may be exposed to what is referred to as chronic early stress, which has no consensual definition, but reflects repeated physical and/or psychological trauma, as well as disturbances in early interactions. In the period of child development when neuroplasticity is important, exposure to chronic early stress can alter the physiological stress response system and thus lead to vulnerability to psychopathology [3].

Several forms of maltreatment are usually identified, such as physical abuse, psychological or emotional abuse or mental cruelty, sexual abuse, and serious neglect or lack of care [4]; they are defined in the table attached in Appendix A. In developed countries (such as the United Kingdom, USA, Australia, Canada, and France), the occurrence of maltreatment has been identified in 1.5% to 5% of children, according to child protection agencies; however, this rate is higher in self-assessment surveys [5]. Children in the care of social services, for example, are much more exposed to negative childhood experiences than the general population; they are significantly more at risk of being subjected (75.5% vs. 33.2%) to a far greater number of negative experiences (2.47 vs. 0.62) [6]. By comparing two meta-analyses that collected international data on mental disorders among young people in the general population and in social services, it was found that children and adolescents in the care of social services have a very high prevalence of mental disorders (49%) [7] compared with 13.4% in the general population [8], i.e., almost four times higher.

Childhood maltreatment increases the risk for psychopathology, both in childhood and adulthood [9], but they may not necessarily correspond precisely to the categorical criteria in the DSM; hence, there is increasing interest in the associated dimensional approach [10,11,12,13]. The phenotypic expression of psychopathology is strongly influenced by exposure to maltreatment, often with earlier onset, more severe symptoms, and a lower response to treatment than in subjects with no history of maltreatment [14].

### Connection with the Hypothalamo–Hypophyseal System

The hypothalamic–pituitary–adrenal (HPA) system is a biological system for regulating stress. In response to stress, the hypothalamus secretes CRH, which stimulates the secretion of ACTH, which in turn stimulates cortisol production in the adrenal glands. The normal circadian rhythm of cortisol secretion reaches a peak 30 min after waking up, followed by a progressive reduction during the day. It would appear that the investigation of the hypothalamic–pituitary axis by repeated measurements of salivary cortisol is a very good approach [15] as this reflects plasma levels very well.

In early development, a child’s hypothalamo–hypophyseal axis is regulated by social interaction; it is immature at birth and thus parental care affects its regulation in the infant. There is normally a reduction in the responsiveness of this axis during the first year of life; however, in the absence of appropriate care, infants register cortisol levels in response to stress that correspond to the levels normally present in younger infants [16]. Furthermore, the basal cortisol level at 15 months of age is inversely associated with the infant’s cognitive development, i.e., the activity of the hypothalamo–hypophyseal axis during early infancy is associated with early cognitive development [17]. An adaptive response in the case of early abuse could thus have long-term effects on how the brain subsequently responds to stress [18].

The repercussions of this chronic early stress had initially been explained by the abnormal activation, in this case, of one of the two types of glucocorticoid receptors, which function in different ways within the brain, and that can lead to deleterious effects, such as epigenetic modifications, effects on brain plasticity, and vulnerability to stress [19]. The results concerning cortisol disturbances in abused children are quite heterogeneous. This is due to the diversity of the populations studied and the differences in the methodology employed in the investigation of the activity of the hypothalamo–hypophyseal axis [10,20,21,22,23]. The results of two recent meta-analyses [24,25] on the links between chronic early stress and impaired cortisol secretion are contradictory; however, one of these meta-analyses included only adult subjects and the other included subjects of all ages. Finally, a meta-analysis published in 2022 found evidence of blunted cortisol stress reactivity in individuals exposed to maltreatment in childhood [26].

Furthermore, numerous studies on the links between maltreatment, disruption of the activity of the hypothalamic–pituitary axis, and psycho-behavioral disorders have been carried out, with contradictory results. These have been difficult to compare because of the diverse populations included and the variety of methodologies employed, for axample, subjects who suffered early maltreatment only [27], those with internalizing disorders [20], those with externalizing disorders [28], those studies that measured the reactivity of the hypothalamic–pituitary axis [29], or those that concentrated on the circadian rhythm [30].

However, to the best of our knowledge, no meta-analysis has specifically investigated the links between cortisol abnormalities and psycho-behavioral disorders in abused children. Therefore, the aim of this systematic review is to provide a clear overview of the most recent findings concerning the link between psycho-behavioral disorders and disruption of the hypothalamic–pituitary system in children who have been maltreated.

We formulated the following hypotheses: subjects with a history of maltreatment are more likely to suffer from psycho-behavioral disorders than subjects who have not been maltreated. These subjects will also have different hypothalamic–pituitary axis activity and these differences in functioning will correlate with psycho-behavioral disorders.

## 2. Materials and Methods

### 2.1. Search Strategy

We carried out a systematic search in June 2020 in the following databases: PubMed, Scopus, Cochrane, and Google Scholar. This review followed the Preferred Reporting Items for Systematic Reviews and Meta-Analyses (PRISMA) guidelines [31] to ensure comprehensive and transparent reporting of the methods and results. The following search equation was used for the various databases mentioned above: (Hypothalamo–Hypophyseal System OR Pituitary–Adrenal System OR hydrocortisone [MeSH Terms]) AND (Adverse Childhood Experiences OR Child Abuse [MeSH Terms]) AND (Behavioral Sciences OR mental disorders OR psychology OR Stress,Psychological [MeSH Terms]) AND (Adolescent OR Child OR Infant [MeSH Terms]) NOT (Adult [MeSH Terms]).

To limit our search to the most recent articles, we chose to filter articles dated from 1 January 2000 to 1 June 2020. Articles in French and English were accepted. All clinical research articles were evaluated: a first selection of articles was completed, after reading the title and abstract, then a second selection, after reading the full article. Studies were included independently of the results. We considered the abstracts of studies that included children and/or adolescents, and that were concerned with the links between maltreatment, psycho-behavioral disorders, and the hypothalamo–hypophyseal axis.

### 2.2. Eligibility Criteria

After the initial selection of abstracts with the search equation in the different databases, studies that met the following criteria were included: (1) original research articles involving children or adolescents aged between 6 and 16 years, that included at least (2) a measurement of the hypothalamo–hypophyseal axis (simple measurement or blood test or salivary cortisol reactivity test), (3) an exploration of the history of the maltreatment (questionnaire, scale, or medico-social records), and (4) an exploration of mental health (psycho-behavioral scale).

These age limits were chosen in an attempt to select studies that used the same psycho-behavioral exploration scales in school-aged children who had access to language. Literature review articles and meta-analyses were not included. Studies on acute trauma, or with a methodology that included structural or functional brain imaging were not included.

### 2.3. Data Collection

The following data were extracted from all of the studies included and grouped in Table 1: population studied, characteristics of the subjects included, control group or otherwise, measurement scales used for mental health, maltreatment, and methodology for investigating the hypothalamo–hypophyseal axis.

The main results and significance thresholds concerning the links between maltreatment, mental disorders, and the hypothalamic–pituitary axis are grouped together in the second table.

## 3. Results

### 3.1. Search Results

A total of 351 abstracts were obtained using the search equations in the various databases cited above. Of these, 35 were selected for further analysis (full reading). Finally, 15 articles, which presented all the eligibility criteria mentioned above, were included in this review.

Details of the search results are summarized in Figure 1.

### 3.2. Characteristics of the Studies

#### 3.2.1. Populations Studied

The number of subjects was 3733 (of which 284 were healthy control subjects).

In the different studies, the subjects were either young children and adolescents or adolescents only, and no study involved only pre-pubertal children. The study populations were heterogeneous: there were subjects linked to social services in most cases [27,29,37,38,41,42,43,44]; in one study, it was children of alcohol-dependent fathers [33], recruited from an addiction center in India (in the study, children with alcohol-dependent mothers were excluded). In another study, the subjects were taken from a longitudinal study (E-Risk study in Great Britain) including twins with a mother aged 20 or younger when they were born [36]. Finally, there were subjects from underprivileged backgrounds [40]; subjects with a psychiatric disorder, such as internalizing disorders [32]; subjects taken from a study on depression in young people [45]; subjects with borderline personality disorder [34]; and subjects with attachment disorder [30]. Two studies included only subjects who had experienced early maltreatment. The mean age of separation from biological parents for one study was 62.9 months (SD 25.3) [30] and for the other study was 1.59 years (SD 1.05) with a period of maltreatment between 0 and 3 years [27]. A third study found an early maltreatment sub-group (maltreatment before the age of 5 years) [37].

Table 1 presents the main characteristics of the subjects included and the different scales used in each study.

#### 3.2.2. Design and Methodology of the Studies

Two of the 15 studies were longitudinal studies: in the first, subjects were children in the general population (in Australia) aged 11–13 years; inclusion was based on the presence of borderline personality disorder. This was followed by analysis of cortisol and of maltreatment in the same subjects when they reached 14–16 years of age [34]. The other subjects were children in social services who had been maltreated, included at an age between 9 and 13 years with cortisol testing at that time, and then mental health questionnaires were carried out, on average, 2.7 years after inclusion [29]. Eleven of the 15 studies had a control group. We labelled the control groups “Group C”. The groups of maltreated children were labelled “Group M”. In Table 1, for studies in which the subjects of the control group were from the same core sample as the maltreated children, we specified, in the “subjects” column, the number of children (*n*) in the total sample and, below, the number of children (*n*) in group M. For studies in which the healthy control group was independent, only the number of children (*n*) in group M was recorded in the “subjects” column.

#### 3.2.3. Maltreatment Assessment

Maltreatment was investigated differently depending on the studies included: social service records, in most cases [27,29,42], mostly using the “maltreatment classification system” (MCS) [37,39,41,43]. In two studies, standardized questionnaires for the foster family were used: the studies that included children who had suffered early maltreatment [30] used these questionnaires, one of which was completed using the social services file [27]. Several studies used questionnaires designed for children: Childhood Trauma Questionnaire (CTQ) [34,35,38,40], Screen for Adolescent Violence Exposure (SAVE) [35], Adverse Childhood Experiences Scale (International Questionnaire) (ACE-IQ) [33], and Childhood Experiences of Violence Questionnaire (CEVQ) [38]. Finally, two other studies used Questionnaires for biological parents: The Early Trauma Inventory by Bremner (Bremner JD et al., 2000) [32], Unspecified Standardized Questionnaire [36], and one study also used, in addition to the child questionnaire, a prenatal stress questionnaire for the mother: the Prenatal Psychosocial Profile [33].

#### 3.2.4. Psycho-Behavioral Assessment

Psycho-behavioral assessment questionnaires, scales, or scores differed between studies, with most studies using more than one. We classified them according to the person completing the scale/questionnaire (child, parent, and educator or teacher) and the type of assessment for child questionnaires (dimensional or categorical). For teachers/educators or parents (dimensional scales), several questionnaires were used, namely the Teacher’s Report Form (TRF), the Behavior Assessment System for Children’s self-report of personality (BASC-SRP), the Child Behavior Checklist (CBCL), and the Strengths and Difficulties Questionnaire, all four designed for behavioral and emotional assessment of children and adolescents. The Relationship Problems Questionnaire (RPQ) was also used, as a questionnaire for assessing the symptoms of attachment disorder.

For categorical assessment, the Mini-International Neuropsychiatric Interview for Children and Adolescents (MINI kid), and the Schedule for Affective Disorders and Schizophrenia for school-aged children (K-SADS-E), both exploring child and adolescent psychiatric disorders (DSM IV), were used. For dimensional assessment, the Children’s Depression Inventory (CDI) (scale for assessing the severity of depressive symptoms), the Adolescent Personality Disorder (APD) (Personality Disorder Symptoms Rating Scale), the Impulsiveness Venturesomeness and Empathy (IVE) Questionnaire (Evaluation of empathy and impulsivity), and the Adolescent Delinquency Questionnaire (ADQ) scales were used. Five out of 15 studies used the TRF scale [36,37,39,41,43]; four studies used the CBCL scale [27,32,35,36] and three studies used the CDI [29,37,41].One study used, in addition to the TRF and the Dissociation Sub-scale, two cognitive assessment scales: California Verbal Learning Test for Children (CVLT-C) and Peabody Picture Vocabulary Test 3 (PPVT3): assessment of memory for verbal comprehension and learning [43]. Several studies used sub-scores or composite scores to more specifically assess interpersonal and social difficulties, better characterize emotional and behavioral problems (particularly fear and aggression) as well as symptomatology (internalizing or externalizing), and measure resilience capacities.

#### 3.2.5. HPA Axis Functioning Assessment

All of the studies investigated the activity of the hypothalamo–hypophyseal axis, as specified in the eligibility criteria. The 15 included studies all used salivary cortisol measurement, either by measuring spontaneous cortisol levels or by measuring cortisol reactivity to induced stress. Seven of the 15 included studies investigated the activity of the hypothalamic–pituitary axis by measuring its reactivity to induced stress: The Trier Social Stress Test for Children (TSST-C) was the test used in the majority of the studies [29,33,35,38,40]. The procedure is as follows: saliva samples are taken to determine pre-test cortisol (basal cortisol), then repeated samples are taken at the time of the test (induced stress: oral presentation in front of a group of examiners) and post-test. However, the number of samples taken differed slightly from one study to another, as did the number of minutes between the test and sampling. It should also be noted that one study out of the six used the calculation of a cortisol–DHEA ratio [35]. One study used the Psychosocial Stress Test, which is similar to TSST-C, but induces stress in a slightly different way from TSST-C [36]. One study used The Socially-Evaluated Cold Pressor Task [32], similar to the TSST-C, which induces a response to heat stress, in addition to social stress. Depending on the study, the response corresponded to the area under the curve (with or without pre-test basal levels), to the slope at the post-stress peak, or to the maximum cortisol level at the post-stress peak.

Eight studies measured spontaneous cortisol levels by taking saliva samples at different times of the day. In three studies, the aim was to measure the circadian rhythm of cortisol [27,30,42], with three samples taken per day, in the morning, at midday, and in the afternoon or at bedtime, following practical sampling instructions to avoid disturbing the salivary cortisol (e.g., no food or toothbrushing just before sampling). One study assessed morning cortisol only [43] and followed practical sampling instructions. One study assessed the reactivity of the hypothalamic–pituitary axis upon awakening [34], using the cortisol awakening response (CAR) protocol with the measurement of cortisol by salivary sampling (3 samples per day over 2 days: upon awakening, and after 30 min and 60 min). However, sampling instructions were not specified in this article. Finally, two studies assessed cortisol at two times of the day: at 09:00 a.m. and at 16:00 p.m. (this is not the circadian rhythm) [37,41], but these studies specified the sampling instructions. Finally, one study investigated the evolution of cortisol levels over time by taking a saliva sample at 16:00 p.m. once a week for 20 weeks [39].

### 3.3. Main Results

We present the main results of the different studies concerning the links between maltreatment, activity of the hypothalamo–hypophyseal axis, and psycho–behavioral disorders. Table 2 presents a summary of these results.

#### 3.3.1. Relationship between Maltreatment and Psycho-Behavioral Disorders

For this sub-section, we differentiated between the results obtained from a dimensional investigation of psycho-behavioral disorders and those obtained from a categorical investigation.

Three out of the 15 studies made no mention of any link between maltreatment and psycho-behavioral disorders.

Four studies have highlighted a correlation between maltreatment and dimensional disorders: positive correlations between CTQ and BPD scores (*p* < 0.001) [34]), between the exposure to violence score (SAVE) and the IS and ES sub-scores of the CBCL (*p* < 0.001) [36], between emotional abuse and IS and unintentional trauma and IS (*p* < 0. 05 for both) [32], and between CTQ and the anger control scores, and a negative correlation between CTQ and interpersonal competence scores (*p* < 0.01 for both) [40]. Eight studies highlighted a significant difference between the maltreated group and the control group in terms of dimensional disorders: a higher dissociation score in group M (*p* < 0.001); higher total difficulty scores and higher RPQ scores in group M (*p* < 0.0001 for both) [30]; a significantly lower resilience score in group M (*p* < 0.001) [41]; a higher total TRF score (*p* < 0.001) and higher ES (*p* < 0.001) and IS (*p* < 0.01) TRF sub-scores in group M [39]; higher impulsivity parts for the CBCL and IVE scores in group M (*p* < 0.001 and *p* = 0.005, respectively) [27]; higher social (*p* < 0.001), emotional (*p* < 0.01), and behavioral (*p* < 0.05) problem scores in group M [36]; and, finally, higher ES sub-score for the SDQ in group M (*p* = 0.003); however, there was no significant difference in the IS sub-score of SDQ (*p* = 0.21) [33].

When considering categorical psychiatric disorders, one study showed that there was no significant difference between group M and the control group in the prevalence of DSM IV psychiatric diagnoses (*p* = 0.34) across all diagnoses [27] (there was no analysis of diagnoses in this study because of the small number of subjects meeting the DSM criteria). Another study, which included only female subjects, found a significant difference in the prevalence of diagnoses of major depressive episodes (MDE) and post-traumatic stress disorder (PTSD) (more frequent in group M, *p* < 0.01, for both diagnoses) [38]. A third study showed a higher prevalence of attention deficit and hyperactivity disorder (ADHD) (*p* = 0.02) and conduct disorder (CD) (*p* = 0.03) in group M using the MINI kid 2, but a similar prevalence of oppositional defiant disorder (ODD) (*p* = 0.77) [33].

#### 3.3.2. Relationship between Maltreatment and the Hypothalamo–Hypophyseal System

First of all, it is interesting to note that some studies pointed out that, even where differences were highlighted between groups, cortisol levels remained within the physiological norm [27,30].

Two of the 15 studies did not mention this association. Three studies did not highlight any association at all: one study, measuring the morning cortisol level, found no significant difference between the two groups [43]; another study found no significant difference between the two groups, either in the levels at different times of the day or in the cortisol circadian rhythm [42]; finally, one study exploring the reactivity of the hypothalamo–hypophyseal axis to induced stress (TSST) found no correlation between the reactivity of the axis and the maltreatment score [40].

In the other two studies that evaluated the circadian rhythm (both involving subjects who had experienced early maltreatment), one concluded that the morning–noon slope of the cortisol curve was shallower (*p* = 0.034) and that the morning cortisol level was significantly lower (*p* = 0.04) in group M (with no difference at the other two times of the day) [27]. The other study found no difference in the rhythm, but a lower morning cortisol level in group M (*p* = 0.047) [30]. Another study showed a negative correlation between the maltreatment score and the morning cortisol level (*p* = 0.02) [41].

Several studies that investigated the reactivity of the hypothalamo–hypophyseal axis highlighted similar results: four studies showed a significant decrease in axis reactivity in group M compared with the control group (*p* < 0.001 [38], *p* < 0.05 [36], and *p* < 0.01) [33], including one in boys only (*p* < 0.01) [29]. Another study found an association between exposure to violence and decreased axis reactivity (*p* = 0.021) [35]. Finally, there was also significantly greater variability in cortisol levels over time in children who had been maltreated than in children who had not been maltreated (*p* < 0.05) [39]. There was also a negative correlation between the CTQ score and cortisol reactivity upon waking (*p* = 0.047); however, when differentiating between the two genders, the correlation was negative for girls and positive for boys.

#### 3.3.3. Relationship between Psycho-Behavioral Disorders and Cortisol Levels

Four studies made no mention of this link. Two studies found no correlation between the disorders and the reactivity of the hypothalamic–pituitary axis to induced stress [29,40].

There was no correlation between the BPD score and cortisol reactivity upon awakening (*p* = 0.537) [34], nor was there any correlation between the cortisol level and CBCL scores (total or sub-scores for either IS or ES) [27]. No correlation was found between cortisol levels and SDQ or RPQ scores [30].

One study exploring the reactivity of the hypothalamo–hypophyseal axis found a negative correlation between the reactivity of the hypothalamo–hypohyseal axis and ES only (*p* = 0.008) (not with IS) [35].

A positive correlation was highlighted between the physical aggression score and the morning cortisol rate/decline, but the correlation was negative for the relational aggression score [42]. There was a negative correlation between the resilience score and the morning cortisol level in the control group only (*p* = 0.05) [41]. There appeared to be a correlation between the TRF score and the variability of cortisol levels over time (*p* < 0.05) [39]. A negative correlation was found in a study of the reactivity of the hypothalamic–pituitary axis (slope at the peak) with IS and ES (*p* < 0.01 for both) [32]. Finally, a correlation was found between the discrimination score and the cortisol level (*p* = 0.004) [43].

#### 3.3.4. Relationship between Psycho-Behavioral Disorders, Cortisol Secretion and Maltreatment

One study made no mention of any results regarding this association [29]. Four studies found no correlation: one investigated circadian rhythm in subjects with an attachment disorder who had experienced early maltreatment [30]; another investigated categorical disorders in girls and found no significant difference in the reactivity of the hypothalamic–pituitary axis between subjects who had a diagnosis of MDE/PTSD and healthy subjects within the maltreatment group [38]; yet another study highlighted several associations between aggression scores and the rate/decline of morning cortisol, which were stronger in the group of children who had not been maltreated [42]; finally, the last study, which observed the variability of cortisol over time, found no correlation between the interaction of cortisol variability/maltreatment and the TRF score (*p* = 0.43) [39].

Several results are in agreement in the studies that investigated the reactivity of the hypothalamic–pituitary axis—in two controlled studies, a negative correlation was highlighted, in group M only, between the reactivity of the axis and the total SDQ score, on the one hand (*p* < 0.05) [33], and the social (*p* < 0.001) and behavioral (*p* < 0.01) problem scores on the other hand [36]. A negative correlation was also found in the interaction of hypothalamic–pituitary axis reactivity and physical abuse with IS (*p* = 0.007) and ES (*p* = 0.013) [32]. Furthermore, the association between exposure to violence and the ES score was thought to be mediated by the reactivity of the hypothalamic–pituitary axis (significant indirect effect with bootstrapping) [35].

There was also a positive correlation between hypothalamic–pituitary axis reactivity and anger control scores in subjects who had suffered a high level of maltreatment (*p* < 0.01) [40]. Finally, in girl only, the interaction between the CTQ score and the BPD score correlated negatively with the cortisol reactivity upon waking (*p* = 0.045) [34].

However, there was a positive correlation regarding the interaction between axis reactivity and emotional abuse with IS (*p* = 0.001) and ES (*p* = 0.005), and a positive correlation for the interaction between axis reactivity and unintentional trauma with IS (*p* = 0.017) and ES (*p* = 0.027) [32].

There were also several concordant results when observing morning cortisol levels: one of the studies that included subjects who had suffered early maltreatment showed a negative correlation between morning cortisol levels and the CBCL score in maltreated children only (*p* = 0.007) (not in the healthy control group) [27].

In addition, children who had been physically abused and who presented a high level of morning cortisol, had a higher resilience score than children who had not been abused (*p* = 0.001) [41].

Physically and/or emotionally neglected children, with low morning cortisol levels, had a higher false recognition score (*p* < 0.001) and a lower discrimination score (*p* < 0.001) compared with subjects who had higher cortisol levels [43].

Finally, it would appear that children who had been maltreated early in life and who had severe depressive or internalized symptoms (high CDI and IS scores in the TRF) had a morning–afternoon cortisol curve with a shallower slope compared with children who had not been maltreated or who were maltreated later in life (*p* = 0.008) [38]).

## 4. Discussion

The aim of this review was to provide clear results regarding the links between mental disorders and alteration of the hypothalamic–pituitary axis (measured by cortisol secretion) in young people (children and adolescents) who had been maltreated. To the best of our knowledge, this is the first systematic review on this subject. We identified 15 studies on this subject that corresponded to our inclusion and exclusion criteria.Several methodological limitations of the studies should be highlighted. Firstly, several studies used a small sample size [27,37,40] and two studies did not use a mixed sample, which limited the generalizability of the results [33,38]. Most studies did not include puberty as a covariate, and one study assessed puberty, but only using a self-administered questionnaire [29]. With regard to the exploration of psycho-behavioral disorders, assessment was sometimes based on self-administered questionnaires, which may have led to measurement bias [35]. Furthermore, the results of subgroups separating different types of maltreatment must be interpreted with caution. Indeed, there is often a co-occurrence of different types of maltreatment within the included populations (entrusted to social services in most studies). Some studies also raised the question of the directionality of the link between HPA axis reactivity and the onset of psychiatric disorders, as there may be a disturbance in cortisol levels in subjects with hetero-aggressive behavior disorders [42].

Furthermore, in the included studies, information concerning prenatal stress was not included in the methodology, which could also be a factor influencing the child’s stress regulation system at a very early stage of development. In addition, the variability of cortisol levels may have been influenced by the different environmental stresses experienced by the child at the time of the study.

Finally, the results are quite contradictory, and it should be stressed that there is significant heterogeneity in the methodology of the studies included, as well as in the samples and scales used. All of these differences between studies make it difficult to compare the results.

### 4.1. The Impact of Maltreatment on the Mental Health of Children and Adolescents

The first interesting result of this work is the confirmation that children who were maltreated had more psycho-behavioral dimensional disorders (internalizing, externalizing, and dissociative symptoms) than children were not maltreated. They were also less resilient. The severity of maltreatment could be correlated with the extent of borderline personality disorder, and the extent of exposure to violence could be correlated with the extent of internalizing and externalizing disorders. In addition, emotional abuse and unintentional trauma could lead to additional internalizing disorders. On the other hand, it would seem that the prevalence of psychiatric disorders (categorical diagnoses) in children who experienced early maltreatment alone was no higher than in children who were not maltreated, whereas, in the same study, dimensional disorders were more significant in the maltreated group [27]. However, this result should be interpreted with caution as it referred to a small sample. On the other hand, it would seem that the diagnoses of MDE and PTSD were more significant in female subjects who were maltreated than in girls who were not maltreated. In addition, diagnoses of ADHD and conduct disorder appeared to be more common among maltreated children. Therefore, it is important to be aware of the possible, different subtypes of the same diagnostic outcome, depending on whether there was a history of maltreatment or otherwise. We referred to these as eco-phenotypes [14,46].

### 4.2. Cortisol Secretion in Maltreated Subjects

It appears that children who experienced early maltreatment had lower morning cortisol levels than children who had not [27,30]. The most common finding was a decrease in the reactivity of the hypothalamic–pituitary axis to induced stress (the slope of the cortisol secretion curve following stress is shallower and/or the peak is lower) in maltreated children compared with children who were not maltreated. This result was consistent with a 2017 meta-analysis that included subjects of all ages [25]. We can interpret this result as an attempt by the brain to adapt to negative childhood experiences [47], resulting in a decrease in the reactivity of the physiological stress system.

Obviously, the genetic aspect must also be considered; it would seem that there was a correlation between certain variants of the CRH1 gene (gene coding for the CRH type 1 receptor) and an alteration in cortisol in maltreated children [48].

### 4.3. Associations between Cortisol Secretion and Psycho-Behavioral Disorders

The results of this review on the links between spontaneous/stress–response cortisol levels and disorders were markedly discordant among the included studies. A 2008 meta-analysis on this topic concluded that there was no relationship between cortisol and externalizing disorders in adolescents, and a weak association between spontaneous cortisol levels and externalizing disorders in children (but with an inverted correlation between pre-school and school-age children) [49].

### 4.4. Associations between Maltreatment, Activity of the Hypothalamo–Hypophyseal Axis, and Psycho-Behavioral Disorders

There was little concordance in these results; the results highlighted by several studies was the decrease in the reactivity of the axis and disorders in the maltreated groups. However, the correlations seemed to be reversed according to the type of maltreatment. We believe these results were too complex to form a definitive conclusion.

When looking at spontaneous cortisol levels, the two results that pointed in the same direction were a lower level of morning cortisol associated with more psycho-behavioral disorders, only in children who experienced chronic early stress, and a higher level of morning cortisol associated with more resilient functioning in maltreated children. However, another study that included children who experienced chronic early stress did not find an association between distress and morning cortisol levels. We believe it is important to study resilient functioning in maltreated children, because identifying differences between sensitive and resilient subjects could help us understand how to promote compensatory brain adaptations [46,50].

It is now accepted that where there has been early maltreatment, there is a modification of cerebral plasticity by molecular brakes that puts an end to certain critical periods of development (including an important role for the GABAergic inhibitory system) [51]. Certain genetic polymorphisms determine cerebral plasticity and, thus, the vulnerability of the brain to stress. This is referred to as phenotypic plasticity [46]. Another possibility is that chronic early stress will increase oxidative stress in the central nervous system and thus increase the risk of developing psychiatric disorders. The involvement of redox mechanisms in physiopathology could represent new therapeutic targets [52]. Finally, the functional imaging alterations demonstrated in maltreated children [53] may also participate in this psychopathological vulnerability, highlighting the need for future research exploring the links between psycho-behavioral disorders, cortisol, and functional imaging data.

### 4.5. Limitations

The results of this literature review must be interpreted with caution due to several limitations. Firstly, there are methodological differences within the studies: in terms of samples (age, subject characteristics, and period and duration of exposure to maltreatment), investigation of disorders, maltreatment, and the hypothalamic–pituitary axis. Furthermore, in addition to the fact that not all of the studies were controlled, some control groups consisted of healthy volunteer subjects who had no connection with social services, and other subjects were only from disadvantaged backgrounds or who had contact with social services but had no history of maltreatment in their medical-social records. In view of all these differences, it was not possible to carry out a meta-analysis of these studies.

Some results should also be interpreted with caution within the different studies in view of the characteristics of the sample such as the possible biases created by the investigation of maltreatment, particularly where the questionnaire targeted the child or the biological parent, and biases caused by the measurement of cortisol secretion taken in the absence of practical sampling instructions. In addition, small sample sizes are sometimes problematic and reduce the power of the studies. Another limitation on the interpretation of all of these results on the functioning of the HPA system is that maltreatment can be confused with contemporary stress in the study, depending on the quality of social support.

Furthermore, this review did not consider genetics, and we know that individuals who have been subjected to maltreatment differ from individuals who have not been maltreated, which may also depend on certain genetic polymorphisms and epigenetic modifications, which may contribute to an increased risk of psychopathology [14]. For example, reviewing the literature has highlighted the involvement of methylation of the NR3C1 gene (epigenetic modification of this gene coding for a glucorticoid receptor) in the predisposition to a major depressive episode in a setting where there is chronic early stress [54]. Epigenetic changes in the glucocorticoid receptor (GR) gene are also thought to be more frequent in children who have been maltreated, and these changes are correlated with the presence of psycho-behavioral disorders [55].

## 5. Conclusions

Despite the various limitations of this review, as outlined above, several results confirmed our hypotheses. Firstly, young people with a history of maltreatment are more likely to present psychiatric disorders than others, which is particularly evident through the dimensional assessments. They also appear less resilient.

With regards to our second hypothesis, the reactivity of the hypothalamic–pituitary axis to stress appears to be diminished in maltreated youth; in particular, morning cortisol levels also appear to decrease in cases of early chronic stress.

As for our third hypothesis, the precise nature and direction of the links between HPA axis dysfunction and psycho-behavioral disorders remain unclear. Further prospective studies involving larger samples are still needed to better understand the role of HPA axis dysregulation in the development of mental disorders in abused children. In particular, longitudinal studies exploring the link between dysregulation of the hypothalamic–pituitary axis and changes in functional imagery in maltreated youngsters would be useful to better understand the neurobiological impact of maltreatment, including genotypical and epigenetic factors.

Indeed, a better understanding of the factors that protect individuals from early adversity is an important challenge for research in this field.

## Figures and Tables

**Figure 1 children-10-01344-f001:**
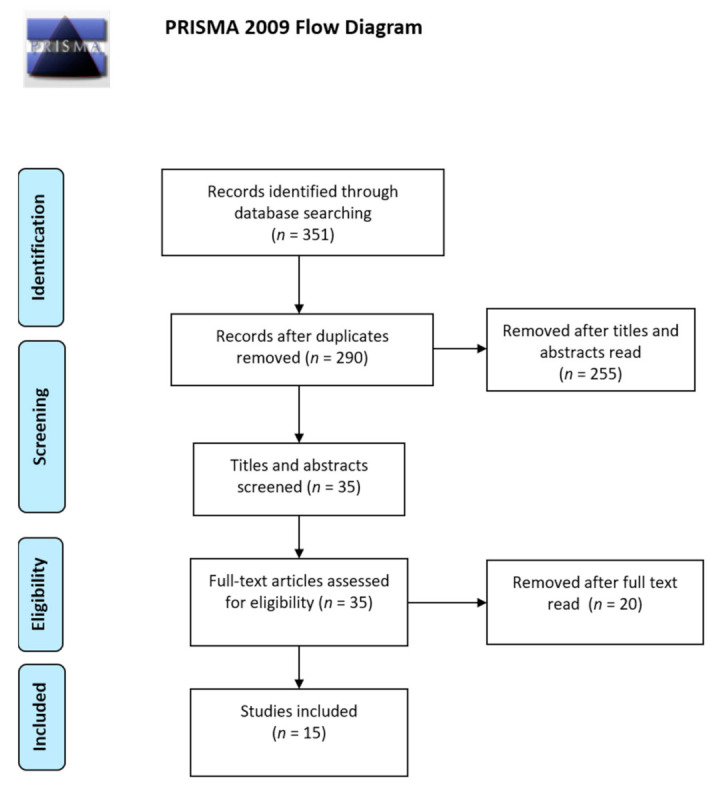
PRISMA flow diagram for the literature search.

**Table 1 children-10-01344-t001:** Characteristics of the different studies.

Authors	Target Population	Subjects (Number, Age)	Control Group (Number, Age)	Maltreatment/Trauma	Cortisol Measurement	Psycho-Behavioral Assessment
Kuhlman et al. [32]	Subjects in a study of depression among young people	*n* = 121 Age 9–16 years Mean age: 12.77 (standard deviation 2.26)	None	Questionnaire “The Early Trauma Inventory”	Socially Evaluated Cold Pressor Task: Measurement of the reactivity to stress of the hypothalamic–pituitary axis	Child Behavior Checklist scale (CBCL)
Timothy et al. [33]	Children of alcohol-dependent fathers	*n* = 50 (male gender only) Age 8–16 years Mean age: 11.3 years (Standard deviation 2.37)	*n* = 50 Of comparable age (*p* = 0.85) and education (*p* = 0.49)	Questionnaires “Adverse Childhood Experiences Scale” (ACE-IQ) and “Prenatal Psychosocial Profile”	Trier Social Stress Test (TSST) Measurement of reactivity to stress of the hypothalamic–pituitary axis	MINI-Kid, Strengths and Difficulties Questionnaire (SDQ)
Kaess et al. [34]	General population, diagnosis of borderline personality disorder	*n* = 69, age 14–16 years, mean Age: 15.5 (Standard deviation 0.4)	None	Questionnaire Childhood Trauma Questionnaire (CTQ)	Measurement of the reactivity of the hypothalamo– pituitary axis on waking (cortisol test, 3 saliva samples/day over 2 days: on waking up, +30 min, +60 min), Instructions for sampling practice not specified	Adolescent Personality Disorder (APD) Scale
Busso et al. [35]	Children from under-privileged backgrounds	*n* = 169 Mean Age: 14.9 (Standard deviation 1.4)	None	Questionnaires CTQ and « Screen for Adolescent Violence Exposure » (SAVE)	Trier Social Stress Test (TSST) Measurement of the reactivity to stress of the hypothalamic–pituitary axis, (cortisol/DHEA ratio)	CBCL
Ouellet- Morin et al. [36]	Subjects in the longitudinal “E-Risk” study	*n* = 190 Age 12 years Group M: *n* = 64	*n* = 126	Standardised questionnaires for parents	Psychosocial Stress Test (PST): Measurement of the Reactivity of the hypothalamic–pituitary axis	CBCL Teachers’ Report Form (TRF), Composite scores for social, emotional and behavioral problems
Cicchetti et al. [37]	Underprivileged backgrounds. M group in connection with social services.	*n* = 493 Age 7–13 Mean age: 10.08 (Standard deviation 1.87) Group M: *n* = 238 (which includes *n* = 51 in Group MP+ for maltreatment after the age of 5 years; and *n* = 187 in Group MP—for maltreatment after age of 5 years	*n* = 255	Exploration of the social services file: scale of maltreatment using the “Maltreatment classification system” (MCS)	Saliva samples for cortisol tests: twice a day for 2 days at 9 am and 4 pm. Practical sampling instructions	Children Depression Inventory (CDI), TRF
Puetz et al. [27]	Children in social care	*n* = 25 Age 8–14 years, mean age: 10.6 (Standard deviation 1.75)	*n* = 26, Mean age: 10.38 (Standard deviation 1.67), comparable in age (*p* = 0.66), IQ (*p* = 0.18) and socio-economic level (*p* = 0.19)	Exploration of social services files and questionnaires for foster families	Assessment of the cortisol circadian rhythm: Saliva samples 3 times/day for 2 days (30 min after getting up, 30 min before lunch and bedtime), Practical instructions for Sampling	CBCL, MINI kid, Impulsiveness venturesomeness Empathy (IVE) Questionnaire
MacMillan et al. [38]	Children in social care	*n* = 67 Age: 12–16 years, Mean age: 14.18 (Standard deviation 1.15) Subjects female gender only	*n* = 25 Mean age 14 (Standard deviation 1.50) comparable in age and gender, but not in socio-economic status (*p* < 0.001)	Questionnaires CTQ and Childhood Experiences of violence Questionnaire (CEVQ)	Trier Social Stress Test (TSST) Measurement of reactivity to stress of the hypothalamic–pituitary axis	Schedule for affective disorders and schizophrenia for school aged children (K-SADS-E)
Doom et al. [39]	Children in social care	*n* = 341 Age 5–13 years Group M: *n* = 187 Average age: 8.4 (Standard deviation 1.8) 3 sub-groups in group M recent, early maltreatment (RE), non-recent, early maltreatment (RE-) and recent, later maltreatment (RL)	*n* = 154	Examination of the social services files: MCS maltreatment system and questionnaires to the families in the control group (maternal-child maltreatment interview)	Measurement of cortisol levels: 1 saliva sample at 4 pm once a week for 20 weeks. Instructions on sampling practices	TRF
Cook et al. [40]	Subjects of a longitudinal study of underprivileged families	*n* = 175 Age 14.5–16 years Average age: 15.36 (Standard deviation 1.01) 2 sub-groups “low maltreatment” for CTQ Score < 32 and “high maltreatment” for CTQ >32 (49% vs. 51%)	None	Questionnaire CTQ	Trier Social Stress Test (TSST): Measurement of the reactivity to stress of the hypothalamic–pituitary axis.	Behavior Assessment Scale for Children Self-report on personality (BASC-SRP), Scores for interpersonal competence and anger management
Cicchetti et al. [41]	Disadvantaged backgrounds. M group in connection with social services	*n* = 677 Age 6–12 years, Mean age: 9.54 (Standard deviation 2.26) Group M: *n* = 347	*n* = 330	Exploring the social services files: scale of MCS maltreatment	Saliva samples twice a day for cortisol tests: at 09:00 a.m. and 16:00 p.m. Practical instructions for sampling	CDI, TRF, Resilience Score
Murray-Close et al. [42]	Disadvantaged backgrounds. M group in connection with social services	*n* = 418 Age 6–12 years Group M: *n* = 219	*n* = 199	Exploration of social services files	Assessment of the cortisol circadian rhythm: Saliva samples 3 times a day: at 09:00 a.m. in the morning, 12:30 p.m. before lunch and 16:00 p.m. Practical instructions for sampling	Scores for physical and relational aggression
Kočovská et al [30]	Early maltreatment and diagnosis of attachment disorder	*n* = 34 Age 5–12 years Average age: 9.4 (standard deviation 1.8) Average age at adoption: 62.9 months (Standard deviation 25.3)	*n* = 32 Average age: 8.7 (Standard deviation 2.4)	Questionnaire standardized for the foster family at the beginning of the placement	Assessment of cortisol circadian rhythm: Saliva samples 3 times a day: 30 min after getting up, before lunch and before bedtime. Practical instructions forsampling	Strengths anddifficulties Questionnaire (SDQ), Relationship Problems Questionnaire (RPQ) Total difficulties score
Cicchetti et al. [43]	Disadvantaged backgrounds. M group in connection with social services	*n* = 317 Age 6–13 years, Mean age: 9.17 (Standard deviation 2.43) Group M: *n* = 143, with 2 sub-groups: physical/emotional (PEN) negligence physical/sexual (PSA) abuse	*n* = 174	Exploration of Social services files: scale of MCS maltreatment	Morning cortisol levels: 1 sample at 09:00 a.m. Practical Instructions for sampling	TRF False recognition and discrimination scores, dissociation sub-scale
Negriff et al. [29]	Children in social care	*n* = 303 Age 9–13 years, Mean age: 10.84 (Standard deviation 1.15)	*n* = 151 Mean age: 11.11 (Standard deviation 1.15)	Exploration of social services files	Trier Social Stress Test (TSST) Measurement of the Reactivity of the axis, Cortisol testing by saliva samples	Adolescent Delinquency Questionnaire (ADQ), CDI

**Table 2 children-10-01344-t002:** Main results on the different associations between maltreatment, cortisol, and psycho-behavioral.

Authors	Links between Maltreatment and Psycho-Behavioral Disorders	Links between Maltreatment and Cortisol	Links between Disturbances and Cortisol	Links: Cortisol-Disorders-Maltreatment
Kuhlman et al. [32]	Positive correlation between emotional abuse and IS and unintentional trauma and IS (*p* < 0.05 for both)	Not stated	Negative correlation of reactivity of the hypothalamic–pituitary axis (slope at peak) with IS and ES (*p* < 0.01 for both)	Negative correlation of hypothalamic–pituitary axis reactivity/physical abuse interaction with IS (*p* = 0.007), and ES (*p* = 0.013), Positive correlation of axis reactivity/emotional abuse interaction with IS (*p* = 0.001), and ES (*p* = 0.005) Positive correlation of axis reactivity/unintentional trauma interaction with IS (*p* = 0.017) and ES (*p* = 0.027)
Timothy et al. [33]	Significant difference between the 2 groups in ES only (increase in group M (*p* = 0.003)	Significant decrease in reactivity in group M (*p* < 0.01)	not stated	Negative correlation between axis reactivity and total SDQ score in group M only (*p* < 0.05)
Kaess et al. [34]	BPD and CTQ score correlation (*p* < 0.001)	Negative correlation between CTQ score and cortisol reactivity on waking up (*p* = 0.047) (However, when differentiating by gender: negative correlation for girls and positive correlation for boys).	No correlation between BPD score and Cortisol reactivity (*p* = 0.537)	Interaction between CTQ score and BPD score correlated with decreased cortisol reactivity on waking in girls only (*p* = 0.045)
Busso et al. [35]	Exposure to violence positively correlates with IS and ES scores (*p* < 0.001 for both).	Exposure to violence negatively correlated with hypothalamo–hypophyseal axis reactivity (*p* = 0.021)	Negative correlation of reactivity of the hypothalamic–pituitary axis with ES (*p* = 0.008) (not with IS)	Association between exposure to violence and ES mediated by the reactivity of the hypothalamo–hypophyseal axis (significant indirect effect with bootstrapping)
Ouellet- Morin et al. [36]	Significant difference between the 2 groups in terms of social (*p* < 0.001), emotional (*p* < 0.01), behavioral (*p* < 0.05) problems (Increase in group M)	Significant decrease in reactivity of the hypothalamo–hypophyseal axis in group M (*p* < 0.05)	Not stated	Significant association between decreased reactivity of the hypothalamo–hypophyseal axis and social (*p* < 0.001) and behavioral (*p* < 0.01) (but not emotional) problem scoresin group M only
Cicchetti et al. [37]	not stated	not stated	not stated	Children in the MP+ group with high IS/CDI scores have a morning-afternoon cortisol slope which is shallower in comparison to groups C and MP (*p* = 0.008)
Puetz et al. [27]	No significant difference in prevalence (MINI kid) between the 2 groups (*p* = 0.34) CBCL (*p* < 0.001) and the impulsivity scores part of the IVE (*p* = 0.005) are increased in group M	Morning–noon slope of the cortisol curve shallower in group M (*p* = 0.034) Morning cortisol significantly lower in group M (*p* = 0.04) (no difference in the other 2 times of day)	No correlation	Negative correlation between morning cortisol level and CBCL score in group M only (*p* = 0.007), No correlation between cortisol slope and CBCL score in either group.
MacMillan et al. [38]	MDD and PTSD prevalence significantly higher in group M (*p* < 0.01 for both)	Significant decrease in reactivity of the hypothalamo–hypophyseal axis in group M (*p* < 0.001)	Not stated	No significant difference in the reactivity of the hypothalamo–hypophyseal axis between subjects with MDE/PTSD and the healthy subjects in the M group. (MDE/PTSD frequency 0% in group C so no comparison possible with this group)
Doom et al. [39]	Total TRF (*p* < 0.001), ES (*p* < 0.001) and IS (*p* < 0.01) scores higher in group M	Cortisol variability over time is significantly greater in group M (*p* < 0.05) Significant difference in cortisol variability between the 3 sub-groups RE, RE- and RL (*p* < 0.05). No difference observed between the sub-groups with different severity of maltreatment	Correlation between TRF score and variability of the level of cortisol (*p* < 0.05)	No correlation between the TRF score and the interaction between cortisol variability and maltreatment (*p* = 0.43)
Cook et al. [40]	Maltreatment positively correlated with anger control score and negatively correlated with interpersonal competence (*p* < 0.01 for both)	No correlation between the reactivity of the hypothalamo–hypophyseal axis and the maltreatment score	No correlation between axis reactivity and anger control and interpersonal competence scores	Positive correlation between axis reactivity with anger control score in the high maltreatment group (*p* < 0.01) and negative correlation in the low maltreatment group (*p* = 0.02)
Cicchetti et al. [41]	Resilience score significantly lower in group M (*p* < 0.001)	No significant difference between the 2 groups but correlation between the maltreatment score and the morning cortisol rate (*p* 0.02)	Negative correlation between resilience score and morning cortisol level only in group C (*p* = 0.05)	In the physical abuse sub-group, and in the presence of a high level of morning cortisol, resilience score significantly higher than in group C (*p* = 0.001) or in sexual abuse sub-group (*p* = 0.021)
Murray-Close et al. [42]	Not stated	No correlation between maltreatment and cortisol rate/rhythm	Positive correlation between physical aggression score and the rate/decline in morning cortisol Negative correlation between relational aggression score and rate/ decline in morning cortisol	No correlation (association between aggression score and cortisol is stronger in Group C than in Group M)
Kočovská et al. [30]	Higher Total Difficulty Score and RPQ score in Group M (*p* < 0.0001 for both)	Similar circadian rhythm between the two groups but lower morning cortisol in group M (*p* = 0.047)	No correlation	No correlation
Cicchetti et al. [43]	Higher dissociation score in group M (*p* < 0.001)	No significant difference in morning cortisol levels between the 2 groups	Correlation between discrimination score and cortisol rate (*p* = 0.04)	NPE sub-group only: subjects with low cortisol levels had higher false recognition scores (*p* < 0.001) and a lower discrimination score (*p* < 0.001) compared with subjects with higher cortisol levels. No correlation shown with subscale dissociation or TRF
Negriff et al. [29]	Not stated	Significant difference in the reactivity of the hypothalamic–pituitary axis between the 2 groups, for the boys only (*p* < 0.01)	No correlation	Not stated

## Data Availability

Not applicable.

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
