# Peer review of "Association between Hpa Axis Functioning and Mental Health in Maltreated Children and Adolescents: A Systematic Literature Review"

_children, 2023, doi:10.3390/children10081344_

Round 1

Reviewer 1 Report

This is a systematic review on the association of childhood adversity with mental disorder and HPA axis dysfunction. The topic is relevant; however, the manuscript lacks basic info about the quality assessment of the studies. The authors need to provide info on quality/bias assessment tool/procedure used in this study.

This manuscript and its figure require English editing by a subject-matter expert.

Author Response

The authors thank the reviewer for his comments.

The authors have added an additional paragraph in the discussion to provide information about the quality assessment of the different studies.

Reviewer 2 Report

This manuscript presents a systematic review of the relationships among child maltreatment, HPA axis dysregulation, and mental health disorder in children.  The study has potential; however, some there are significant concerns.  The specific research questions are unclear as is the unique contribution of the review to the literature, given what seems to be several published reviews on the topic.  The methods are incomplete or unclear - for example, only search terms for one of the 4 databases are provided (2.1).  In addition, in 2.2 it seems inconsistent with the results reported that only clinical trials were eligible.  The results narrative needs significant development (this section reads as unfinished). Rather than bullet points, cogent paragraphs are needed to complement information in the tables. Finally, conclusions should be much more specific and reflect greater synthesis of the overall findings and key conclusions of the study.  

Author Response

The authors would like to thank the reviewer for his comments.

With regard to our specific research question, additional elements have been added to clarify the specificity and contribution of this literature review. Indeed, to our knowledge, no meta-analysis has specifically studied the links between cortisol abnormalities and psychobehavioral disorders.

 In addition, the search terms of the four databases were specified. 

In addition, we would like to thank the reviewer for his vigilance, and the statement that only clinical trials were eligible has been removed.

Finally, the presentation of results has been completely revised, with bullet points replaced by paragraphs.

Reviewer 3 Report

Well written sytematic literature review regarding the relation between the stress-system (HPA axis) and mental health in maltreated children and adolescents.

Adequate methodology.
Results are well written and discussion is well worked out

Author Response

The authors sincerely thank the reviewer for his careful proofreading and positive comments.

Round 2

Reviewer 1 Report

The authors have addressed the issues raised by the reviewers.

Author Response

We thank the reviewer for his comments which helped improve the article. A graphical abstract has been added at the editor's request.

Reviewer 2 Report

Thank you for your revisions to this paper.  This is much improved. I have a few remaining concerns for the results section.

1)  In Table 1, can the authors clarify the reporter for measures of maltreatment and the psychobehavioral assessment?  In addition, some of the measures are ambiguous (e.g., Murray-Close  psychobehavioral assessment; Kočovská child maltreatment measure).  Can you clarify whether these non-standardized forms?  

2) To table 2, please add parameter estimates and either standard errors or confidence intervals. 

3) I found the use of acronyms excessive and to make this section difficult to follow.  Please reduce this.

4) To further improve readability of the results, please further integrate rather than list findings. Paragraphs should have a minimum of 3 sentences. Beginning each section with a summary sentence about how many studies examined the relationship and how many found a relationship would help orient the reader. Perhaps a summary sentence could be added at the end of each section. 

5) The language "no mention" and "not highlight" makes it ambiguous whether the relationship was examined, or whether it was examined but not reported.  Please clarify. 

6) Please consistently cite studies in text as you report their findings (also in the Discussion). 

7) Please add to the beginning of the last section of results what relationships were studied ("this association" and "no correlation" are vague). Clarify how this differs from the prior section.  

8) I have one comment about the discussion.  A sentence seems to be missing at the end of 4.3 (What does the study add to the prior meta-analysis?). 

There are some very minor issues.  In addition to what is noted above I would consider rephrasing "it would seem" language in the discussion. 

Author Response

(The authors gave the same response as above.)
